# Bacterial flagella grow through an injection-diffusion mechanism

Thibaud T Renault[1,2], Anthony O Abraham[3], Tobias Bergmiller[4], Guillaume Paradis[5], Simon Rainville[5], Emmanuelle Charpentier[2], Călin C Guet[4], Yuhai Tu[6]*, Keiichi Namba[3,7]*, James P Keener[8]*, Tohru Minamino[3]*, Marc Erhardt[1]*

[1]Junior Research Group, Infection Biology of *Salmonella*, Helmholtz Centre for Infection Research, Braunschweig, Germany; [2]Max Planck Institute for Infection Biology, Berlin, Germany; [3]Graduate School of Frontier Biosciences, Osaka University, Osaka, Japan; [4]Institute of Science and Technology Austria, Klosterneuburg, Austria; [5]Department of Physics, Engineering Physics and Optics, Laval University, Quebec City, Quebec, Canada; [6]IBM Thomas J Watson Research Center, New York, United States; [7]RIKEN Quantitative Biology Center, Suita, Japan; [8]Department of Mathematics, University of Utah, Salt Lake City, United States

**\*For correspondence:** yuhai@us. ibm.com (YT); keiichi@fbs.osaka-u. ac.jp (KN); keener@math.utah.edu (JPK); tohru@fbs.osaka-u.ac.jp (TM); marc.erhardt@helmholtz-hzi. de (ME)

**Competing interests:** The authors declare that no competing interests exist.

**Abstract** The bacterial flagellum is a self-assembling nanomachine. The external flagellar filament, several times longer than a bacterial cell body, is made of a few tens of thousands subunits of a single protein: flagellin. A fundamental problem concerns the molecular mechanism of how the flagellum grows outside the cell, where no discernible energy source is available. Here, we monitored the dynamic assembly of individual flagella using in situ labelling and real-time immunostaining of elongating flagellar filaments. We report that the rate of flagellum growth, initially ~1,700 amino acids per second, decreases with length and that the previously proposed chain mechanism does not contribute to the filament elongation dynamics. Inhibition of the proton motive force-dependent export apparatus revealed a major contribution of substrate injection in driving filament elongation. The combination of experimental and mathematical evidence demonstrates that a simple, injection-diffusion mechanism controls bacterial flagella growth outside the cell.

## Introduction

Many bacteria move by rotation of a helical organelle, the flagellum. The external flagellar filament is several times longer than a bacterial cell body and is made out of up to 20,000 flagellin subunits (*Berg and Anderson, 1973*; *Chevance and Hughes, 2008*; *Macnab, 2003*; *Silverman and Simon, 1974*) (*Figure 1a*). A type III export apparatus located at the base of the flagellum utilizes the proton motive force (pmf) as the primary energy source to translocate axial components of the flagellum across the inner membrane (*Minamino and Namba, 2008*; *Paul et al., 2008*; *Minamino et al., 2011*; *Erhardt et al., 2014*). Exported substrates travel through a narrow 2 nm channel within the structure and self-assemble at the tip of the growing flagellum. It has been a mystery how bacteria manage to self-assemble several tens of thousands protein subunits outside the cell, where no discernible energy source is available. Previous reports in the literature concerning the mechanism of flagellum growth have been conflicting (*Iino, 1974*; *Aizawa and Kubori, 1998*; *Turner et al., 2012*; *Evans et al., 2013*). An exponential decay of filament elongation with length was observed using electron microscopic measurements, which was proposed to be a result of decreased translocation

**eLife digest** Most bacteria are able to move in a directed manner towards nutrients or other locations of interest. Many move by rotating long tail-like filaments called flagella that stick out from the cell. The flagellum is a remarkably complex nanomachine. It is several times longer than the main body of the bacterial cell body and its external filament is made of thousands of building blocks of a single protein called flagellin. This protein is made inside the cell and a structure at the base of the flagellum known as a type III secretion system uses chemical energy to pump it out of the cell so that it can be incorporated into the growing flagellum. The exported building blocks travel through a narrow channel within the flagellum and self-assemble at the tip.

It has been a mystery for several decades how bacteria manage to assemble the building blocks of flagella outside of the cell, where no discernible energy source is available. Renault et al. used mathematical modeling, biochemical and microscopy techniques to observe how the flagella of a bacterium called *Salmonella enterica* assemble in real time. The experiments demonstrate that simple biophysical principles regulate the assembly of the flagellum. The building blocks are pumped into the channel of the flagellum by the type III secretion system and then diffuse to the tip of the filament. Accordingly, the longer the flagellum gets, the slower it grows. This molecular mechanism also explains why the growth of bacterial flagella will eventually stop even without any other control mechanisms in place.

Further work will be needed to understand how the type III secretion system harnesses chemical energy to drive the movement of flagellin out of the cell into the growing flagellum. A molecular understanding of these processes will aid the design of new antibiotics targeted against type III secretion systems.

efficiency (*Iino, 1974*; *Tanner et al., 2011*). A recent study used dual-colour fluorescent labelling of flagellar filaments to distinguish basal from apical filament growth and found that the rate of polymerization was independent of filament length (*Turner et al., 2012*; *Stern and Berg, 2013*). A model based on the pulling force of a filament-spanning chain of flagellin subunits was proposed to explain the apparent length-independent growth (*Evans et al., 2013*).

## Results and discussion

### Enhanced flagellin export in the absence of assembled filament

In order to test whether filament length itself affects the export rate of flagellin subunits during filament formation, we constructed a flagella-assembly mutant deleted for the first hook-filament junction protein (Δ*flgK*). This resulted in direct secretion of flagellin monomers into the culture media without transport through the elongated filament. The total amount of extracellular flagellin was analysed in the wild-type and the Δ*flgK* mutant by de-polymerizing flagellar filaments into flagellin monomers using heat treatment at 65°C. The amount of extracellular flagellin was approximately 1.6-fold higher in the Δ*flgK* mutant compared to wild-type cells. Consistently, cytoplasmic flagellin was substantially more abundant in the wild-type than in the Δ*flgK* mutant (*Figure 1b*). Measurements of flagellin leakage during filament formation revealed that only a small fraction of the total flagellin is leaked in monomeric form by wild-type cells during filament formation (*Figure 1—figure supplement 1*), demonstrating that the majority of exported flagellin subunits are incorporated into the growing filament under our experimental conditions. These results indicate that the presence of an assembled filament decreases the rate of flagellin transport, which is consistent with the decreased rates of FlgE and FliK export in a long hook mutant (*Koroyasu et al., 1998*; *Erhardt et al., 2011*). A similar filament length-dependent effect on flagellin transport was also observed in a mutant of the flagellin-specific cytoplasmic chaperone FliS (*Figure 1b*). FliS promotes docking and subsequent unfolding of flagellin at the export apparatus (*Kinoshita et al., 2013*; *Furukawa et al., 2016*), suggesting that the flagellin injection rate at the export apparatus substantially contributes to the flagellum growth dynamics.

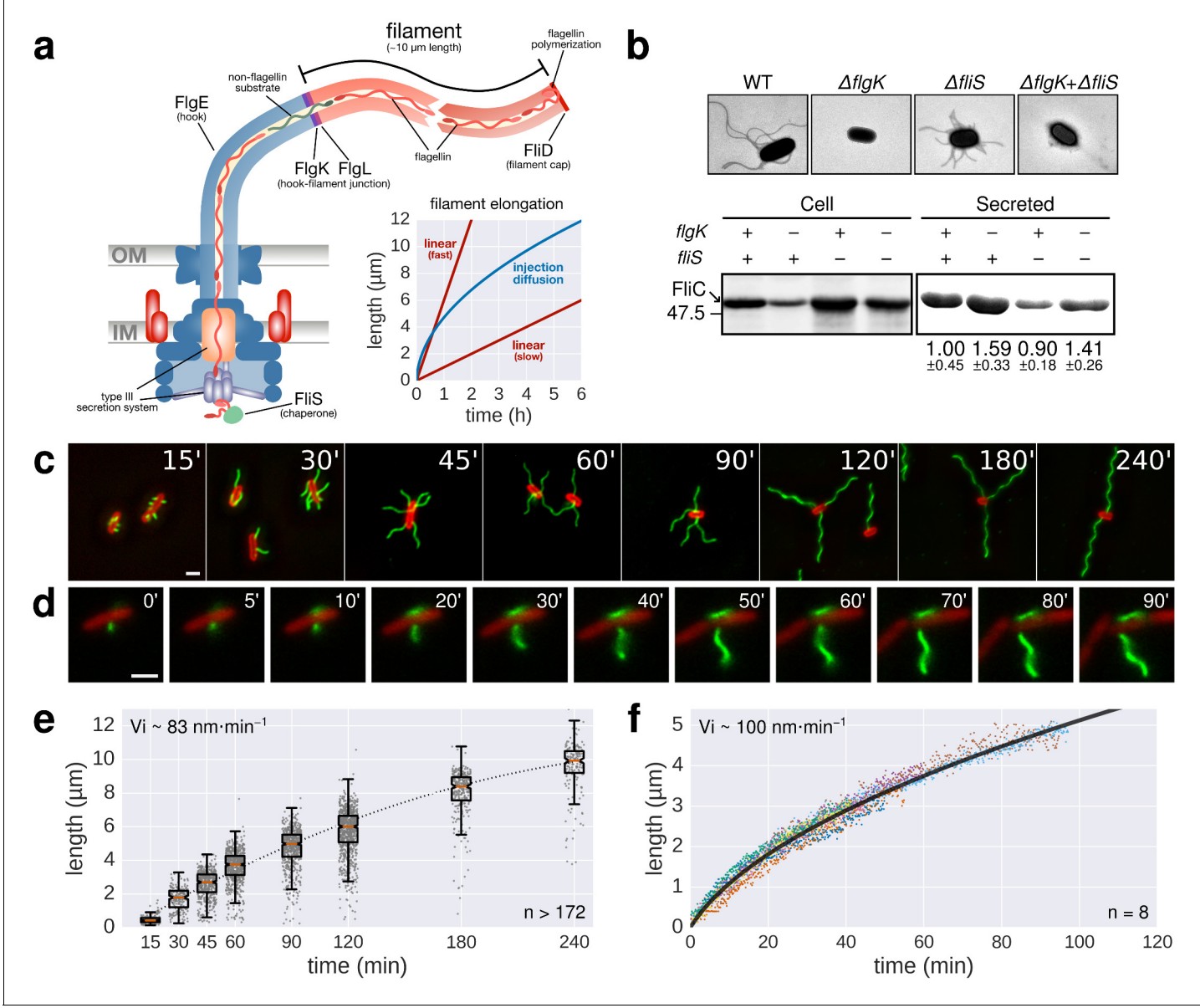

**Figure 1.** Flagellin protein export and flagella growth rate decrease with filament length. (**a**) Schematic depiction of the bacterial flagellum and proposed models to explain the filament elongation dynamics (*Iino, 1974*; *Turner et al., 2012*; *Evans et al., 2013*). OM=outer membrane, IM=inner membrane. (**b**) Top: Electron micrograph images of mutants deficient in the hook-filament junction protein FlgK or the flagellin-specific chaperone FliS. Bottom: Immunoblotting of cellular and Coomassie-staining of secreted flagellin (FliC) in ΔflgK and ΔfliS mutant strains (relative secreted flagellin levels report mean ± s.d., $n = 3$). (**c**) Representative images of a population-based flagellin immunostaining time-course. Time in minutes after induction of flagellin synthesis is indicated. (**d**) Continuous in situ flagellin immunostaining reveals elongation kinetics of individual filaments in real time. Exemplary movie frames are shown and elapsed time in minutes after start of imaging is indicated. (**e**) Quantification of the population immunostaining. Measured filaments per group: $t_{15'}$ ($n = 187$), $t_{30'}$ ($n = 206$), $t_{45'}$ ($n = 480$), $t_{60'}$ ($n = 648$), $t_{90'}$ ($n = 700$), $t_{120'}$ ($n = 827$), $t_{180'}$ ($n = 302$), $t_{240'}$ ($n = 172$). The box plot reports the median (in red), the 25th and 75th quartiles and the 1.5 interquartile range. (**f**) Quantification of the continuous in situ flagellin immunostaining. The dark line represents the global, averaged fit of 8 individual filaments. Raw data shown as coloured dots excluding measurement incidents as explained in *Figure 1—figure supplement 2*. The initial velocity (Vi) was measured on the initial, linear part of the growth curve. Scale bars 2 µm.

The following figure supplements are available for figure 1:

**Figure supplement 1.** Quantitative measurements of flagellin leakage during filament formation.

**Figure supplement 2.** Growth of individual filaments monitored by continuous flow real-time immunostaining.

# The elongation rate of bacterial flagella inversely correlates with filament length

We next measured the growth kinetics of flagellar filaments to determine the relation between growth rate and filament length. We engineered a *Salmonella* strain where the production of flagellar basal bodies (using the *flhDC* flagellar master regulatory operon under control of a anhydrotetracycline inducible promoter) is uncoupled from the expression of chromosomally-encoded flagellin (using the flagellin gene *fliC* under control of an arabinose inducible promoter). This well-established setup allowed for synchronization of flagella production (*Erhardt et al., 2011*; *Karlinsey et al., 2000*) by first assembling basal bodies before initiating filament synthesis. The flagella of the synchronized culture were immunostained after increasing growth times (*Figure 1c*). The initial filament growth rate was ~83 nm·min$^{-1}$, which decreased over time (*Figure 1e*). In a complementary approach, we monitored, in real-time, the dynamic assembly of individual filaments by employing a continuous in situ immunostaining approach (*Berk et al., 2012*) to visualize growing flagella (*Figure 1d*, *Video 1*). A *Salmonella* strain harbouring a functional, hemagglutinin-epitope tagged flagellin variant under its physiological promoter was grown in a microfluidic device in the presence of labelled, primary antibodies. We observed an initial filament growth rate of ~100 nm·min$^{-1}$, which decreased over time similar as for the population-wide approach described above (*Figure 1f*, *Figure 1—figure supplement 2*).

In a previous study, *Turner et al. (2012)* measured the growth kinetics of individual filaments in *Escherichia coli* by site-specific labelling of flagellin subunits containing an exposed cysteine residue using sulfhydryl-specific (maleimide) fluorochromes and reported a length independent growth rate of ~13 nm·min$^{-1}$. We optimized this method to exchange dyes multiple (three to six) times in situ during normal culture growth with minimal perturbation of bacterial growth (*Figure 2*, *Figure 2—figure supplement 1*, *Figure 2—figure supplement 2*, *Figure 3*, *Figure 3—figure supplement 1*). The labelling of successive fragments of the flagellum with maleimide fluorochromes in situ allows observation of the filament growth dynamics at the end of the experiment. Triple labelling (exchange of dyes three times) demonstrated that the extension length of the filament (apical fragment) is inversely proportional to its initial length (basal fragment), until the growth rate for long filaments decreases to a point where it becomes minimal (*Figure 2*). Using this setup, the dynamic range of basal fragment lengths was increased by performing the experiment with varying growth durations (15 to 180 min).

Next, multiple labelling (exchange of dyes six times) of flagellar filaments allowed us to compute various basal/apical couples and increased the dynamic range of the growth rate data for individual flagella. The multiple labelling of flagellar filaments confirmed the length-dependent elongation mechanism with an elongation speed decreasing gradually from ~100 nm·min$^{-1}$ to ~20 nm·min$^{-1}$ (*Figure 3*, *Figure 3—figure supplement 1*). Alternative combination of the fragments allowed us to determine the filament elongation kinetics for various growth durations and *in fine* to derive a growth curve (*Figure 3c–d*). Our method further allowed us to exclude stalled or broken filaments and study the filament elongation dynamics under normal cultivation conditions for a wide range of fragment lengths. We note that we only observed a minor fraction of flagella that broke or stopped growing during the experiment (*Figure 3e*).

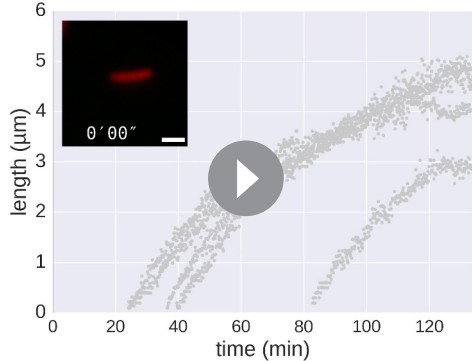

**Video 1.** Real-time flagellum growth observed using in situ continuous flow immunostaining. The animation represents the raw data of the filament length measurements of five representative flagella as a function of time. The inset depicts a 400× time-lapse movie of the corresponding microcolony grown in a CellASIC microfluidic device in the presence of 10 nM anti-HA fluorochrome-coupled primary antibodies. Elapsed time is depicted in min'sec''. Coloured circles highlight the onset of filament assembly of the respective length measurement data. Arrows denote growth or measurement incidents (*e.g.* filament flipped out of focus or broke off). Scale bar 1 μm.

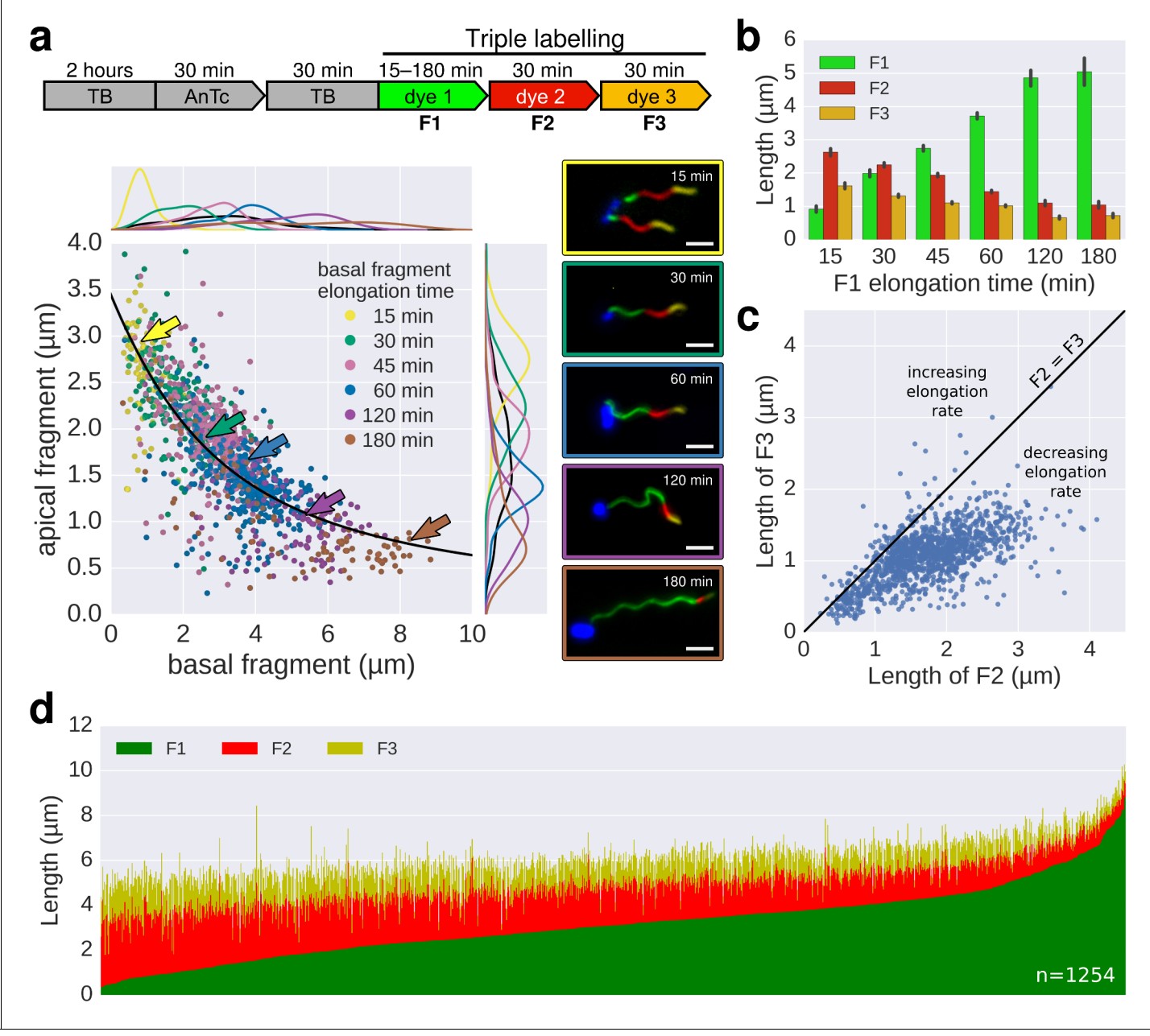

**Figure 2.** In situ filament labelling reveals a negative correlation between filament length and elongation rate. (a) Experimental design of the in situ triple-colour labelling time-course. Basal (F1) and apical (F2) fragments were grown for 15–180 min and 30 min, respectively. The growth duration of basal fragments is indicated in the legend. Coloured arrows indicate the coordinates of the representative example images. The fit represents the injection-diffusion model with parameters $k_{on} \approx 33.35\,\mathrm{s}^{-1}$ and $D \approx 5.90 \times 10^{-13}\,\mathrm{m}^2 \cdot \mathrm{s}^{-1}$. Scale bar 2 μm. (b) Average size of the individual fragments for different durations of elongation of the first fragment. Error bars represent the 95% confidence interval of mean estimation. (c) Relation between the size of the second and third fragment. 93.4% of the filaments have F3 fragments shorter than the F2 fragment with the difference increasing with the length of F2. (d) Flagella labelled in panel a were measured and sorted according to the length of F1, which reveals the inverted relationship between the initial length and extension length of the filament. Each vertical line represents an individual filament ($n = 1254$).

The following figure supplements are available for figure 2:

**Figure supplement 1.** In situ labelling of flagella using maleimide fluorochromes.

**Figure supplement 2.** Triple-colour labelling time course of second fragment F2.

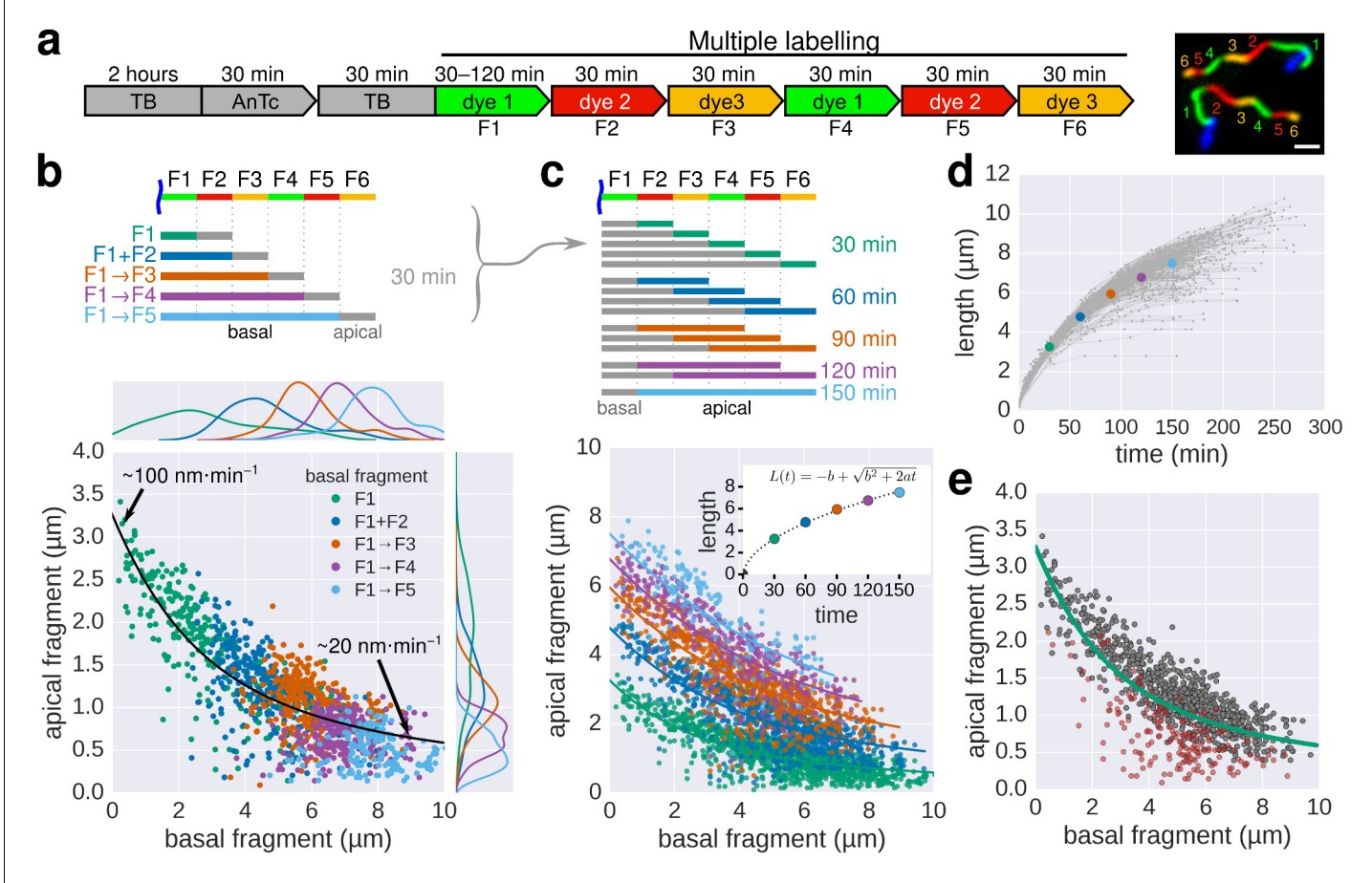

**Figure 3.** Growth kinetics of individual flagella revealed by in situ, multicolour labelling. (a) Left: Experimental design of the in situ, multicolour labelling. Right: Representative fluorescent microscopy image for multiple labelling of flagellar filaments with a series of maleimide dyes. TB: tryptone broth without dye, AnTc: anhydrotetracyline induction of flagella genes. Scale bar 2 μm. (b) Basal/apical length coordinates were obtained by varying the duration of basal growth and successive fragments were combined to generate a total of 1276 basal/apical coordinates from 291 filaments. The growth duration of the apical fragment was 30 min. Average speeds are calculated from the average elongation per 30 min (<1 μm or >8 μm). The fit represents the injection-diffusion model with parameters $k_{on} \approx 27.09$ s$^{-1}$ and $D \approx 5.41 \times 10^{-13}$ m$^2 \cdot$ s$^{-1}$. (c) Basal/apical length coordinates were obtained for various durations of apical growth (30–150 min) from the multiple labelling data shown in panel b. ($n = 1276$ for 30 min, $n = 986$ for 60 min, $n = 697$ for 90 min, $n = 422$ for 120 min, $n = 169$ for 150 min). The fit for various durations of apical growth represents the injection-diffusion model with parameters $k_{on}$ and $D$ (60 min: $k_{on} \approx 27.72$ s$^{-1}$, $D \approx 5.56 \times 10^{-13}$ m$^2 \cdot$ s$^{-1}$; 90 min: $k_{on} \approx 28.06$ s$^{-1}$, $D \approx 5.63 \times 10^{-13}$ m$^2 \cdot$ s$^{-1}$; 120 min: $k_{on} \approx 27.03$ s$^{-1}$, $D \approx 5.42 \times 10^{-13}$ m$^2 \cdot$ s$^{-1}$; 150 min: $k_{on} \approx 26.36$ s$^{-1}$, $D \approx 5.29 \times 10^{-13}$ m$^2 \cdot$ s$^{-1}$). Average growth rates were estimated from the Y-intercept of the fit curve. The inset presents the average growth plotted against time. (d) Filament length as a function of time of individual flagella from the multiple labelling data. Each grey line represents the growth curve of an individual filament. The average growth rates estimated in panel c are plotted for comparison. (e) Quality of multiple labelling data. Only a minor fraction of the filaments were broken or stalled (highlighted as red dots, *Figure 3—figure supplement 1a*), which has limited effect on the parameter fit.

The following figure supplements are available for figure 3:

**Figure supplement 1.** Quality of multiple labelling data.

**Figure supplement 2.** Filament breaking/stalling events and heterogeneous injection rates decrease the quality of the data required to fit the injection-diffusion model.

## An injection-diffusion mechanism explains the growth dynamics of flagellar filaments

The solid curves in *Figure 2* and *Figure 3* represent the best fit of the data to a growth curve for which the growth rate is a function of the length $L$ of the form $\frac{a}{b+L}$, where the parameter $a$

has units of a diffusion coefficient, and $b$ has units of length. Derivation of this formula is based on an injection-diffusion model where flagellin monomers, which are at least partially $\alpha$-helical inside the channel (*Shibata et al., 2007*), are pushed by a pmf-driven export apparatus into the channel and move diffusively in one dimension through the length of the flagellum (*Stern and Berg, 2013*; *Keener, 2006*). An analytical expression for the flagellum length dependent growth rate is based on a continuum injection-diffusion model for the growth of flagellar filaments. Monomers (each of length $l$) in the growing filament are assumed to move diffusively. Because the filaments are long narrow tubes, monomers are partially unfolded and diffusion is constrained to be strictly one-dimensional, *i.e.* no passing allowed. In the corresponding continuum model, we define $\frac{u(x,t)}{l}$ as the density of monomers per unit length at position $x$ and time $t$. Then $u$ satisfies the diffusion equation

$$u_t = Du_{xx}. \tag{1}$$

Here, $D$ is the diffusion coefficient of the monomers. We assume that all end-to-end collisions between monomers are ballistic, with no end-to-end binding. For this, Fickian diffusion is the appropriate description of diffusion, even at high densities. We assume that at the growing end $X = L$, monomers are quickly removed by folding/polymerization so that effectively $u(L,t) = 0$.

The details of the mechanism by which monomers are secreted at the basal end $X = 0$ is not known, but it is known to be related to the pmf (*Paul et al., 2008*). We assume that the rate of secretion (number of monomers per unit time) into an empty filament is $K_{on}$, but if it is not empty, then the rate of secretion is decreased because of the no-passing restriction. Consequently, the flux $J_0$ (number of monomers per unit time at the basal end) is taken to be

$$J_0 = \frac{-D}{l}u_x(0,t) = K_{on}\left(1 - u(0,t)\right). \tag{2}$$

Finally, the rate of growth of the filament is given by

$$\frac{dL}{dt} = \beta J_L = \frac{-D\beta}{l}u_x(L,t), \tag{3}$$

where $\beta$ is the length increment of the filament due to polymerization of a single monomer.

Since the filament growth rate is small compared to the average velocity of monomers, it is reasonable to take the monomer diffusion to be in quasisteady state, *i.e.* $u_{xx} = 0$. Thus, the monomer density in the filament is a linearly decreasing function and $u_x$ is the constant $\frac{-u(0)}{L}$. It follows that the filament growth rate is

$$\frac{dL}{dt} = \frac{\beta D}{l}\frac{1}{\frac{D}{k_{on}l} + L} = \frac{a}{b + L}, \tag{4}$$

where $a = \frac{\beta D}{l}$, with units of diffusion, and $b = \frac{D}{k_{on}l}$, with units of length. This is readily solved to find the filament length as a function of time

$$L(t) = -b + \sqrt{b^2 + 2at}. \tag{5}$$

We can estimate the diffusion coefficient using $a = \frac{\beta D}{l}$, so that

$$D = \frac{al}{\beta}. \tag{6}$$

From all the datasets presented above, we determined $a \approx 0.2\ \mu m^2 \cdot min^{-1}$. Using $\beta = 0.47$ nm (a flagellar filament of 1 $\mu$m length is composed of approximately 2130 flagellin subunits [*Yonekura et al., 2003*]), $l = 74$ nm (assuming an extended, $\alpha$-helical flagellin molecule) this leads to an estimate of $D \approx 5.25 \times 10^{-13}\ m^2 \cdot s^{-1}$. Stern and Berg (*Stern and Berg, 2013*) estimated $D \approx 5.78 \times 10^{-11}\ m^2 \cdot s^{-1}$ for freely moving $\alpha$-helical flagellin in water. The actual diffusion coefficient for movement in the narrow 2 nm channel would be substantially smaller, however. Stern and Berg (*Stern and Berg, 2013*) (their *Figure 2*) used a 480 times smaller diffusion coefficient ($D \approx$

$1.25 \times 10^{-13}$ m$^2 \cdot$ s$^{-1}$) for numerical simulations that resulted in a declining growth curve, which closely resembled the filament growth kinetics presented above.

Our triple and multiple labelling experiments demonstrated that the growth of a new part of the filament (apical fragment) shows a strong inverse dependence on its initial length (basal fragment) for short filaments, while the growth rate for long filaments decreases to a point where this dependence becomes minimal (*Figure 2*, *Figure 3*, *Figure 3—figure supplement 1*). We note that several differences in the experimental setup of *Turner et al. (2012)* from ours might have affected the injection rate and frequency of filament breakage. As described in detail in Appendix 1, the possibility of broken/stalled filaments and possible perturbations of the injection rate reconcile our data with the reported filament growth data of *Turner et al. (2012)* and explains why we observed a length-dependent decrease in growth rate. In support, we simulated in *Figure 3—figure supplement 2* the effects of filament breaking/stalling events and heterogeneous injection rates. The simulated broken/stalled filaments accumulate on the x-axis, which results in a quasi-linear fit of the complete filament growth rate data, similar to the linear filament growth observed by *Turner et al. (2012)*.

We further note that a length-dependent decrease in filament growth rate would explain why flagellar filaments do not growth indefinitely. However, flagellar filaments broken by mechanical shearing forces can re-grow (*Turner et al., 2012*; *Rosu and Hughes, 2006*; *Vogler et al., 1991*). The injection-diffusion model predicts that the elongation rate of re-growing filaments would increase compared to unbroken filaments. We performed multiple labelling in situ to determine the growth rate of individual filaments that had been broken using mechanical shearing forces. Consistent with the injection-diffusion mechanism, the elongation rate of re-growing, previously broken filaments was substantially faster than the elongation rate of unbroken filaments and was dependent on the length of the basal filament segment, which remained attached to the bacterial cell surface (*Figure 4*).

## Inter-subunit chain formation does not contribute to flagella growth dynamics

Based on the observations of *Turner et al. (2012)*, *Evans et al. (2013)* developed a model where folding of newly arriving subunits at the tip of the flagellum would provide energy to pull successive subunits through the channel at a constant rate. Evans *et al.* demonstrated that N-terminal regions of flagellar substrates (FlgD, FlgE, FlgG and FliK) can bind to the C-terminal cytoplasmic domain of FlhB, which is a component of the pmf-driven transmembrane export gate complex. Further, site-specific cysteine-cysteine crosslinking showed that the N- and C-terminal regions of hook (FlgE) and flagellin (FliC) can interact to form head-to-tail dimers. They hypothesized that formation of inter-subunit chains resulting from those interactions could enable their transport at a length-independent speed, as the folding of the exported molecules at the filament tip would provide a continuous pulling force. While the N- and C-terminal interactions of flagellar substrates might play an important role during substrate docking and in the final fold of assembled hook and filament subunits, the proposed inter-subunit chain mechanism for flagellin transport and filament assembly raises several issues that are incompatible with the known biophysical properties of flagellum assembly (*Yonekura et al., 2003*; *Samatey et al., 2001*). A flagellum-spanning chain requires interactions of the N- and C-terminal α-helical domains of flagellin, but the 2 nm wide filament channel (*Yonekura et al., 2003*) is too narrow to accommodate the secretion of much more than one folded α-helix (*Figure 5a*). The chain mechanism hypothesizes that folding of a flagellin subunit at the tip of the flagellum can pull a chain of succeeding subunits, but the N- and C-termini of successive flagellin molecules are anti-parallel and far apart in the polymerized filament structure (~17 Å on average) (*Yonekura et al., 2003*; *Samatey et al., 2001*) (*Figure 5b*). Further, the chain mechanism is not compatible with simultaneous secretion of non-chaining substrates (*Figure 5d*). Flagellar substrates such as FlgM or excess hook-associated proteins (FlgK, FlgL, FliD) are constantly exported during flagellum growth (*Komoriya et al., 1999*) and do not interact with flagellin (*Furukawa et al., 2002*). Also, premature termination of translation is occurring frequently (~$1 \cdot 10^{-4}$ to ~$5 \cdot 10^{-4}$ events per codon) (*Sin et al., 2016*). Thus, a high proportion of 5–20% newly synthesized flagellin might be truncated for the C-terminal domain needed for head-to-tail chain formation. We estimate that secretion of as little as one non-chaining substrate every 10,000 full-length flagellin molecules would prevent filament elongation if a chain mechanism drives flagellum growth (*Figure 5d–g*).

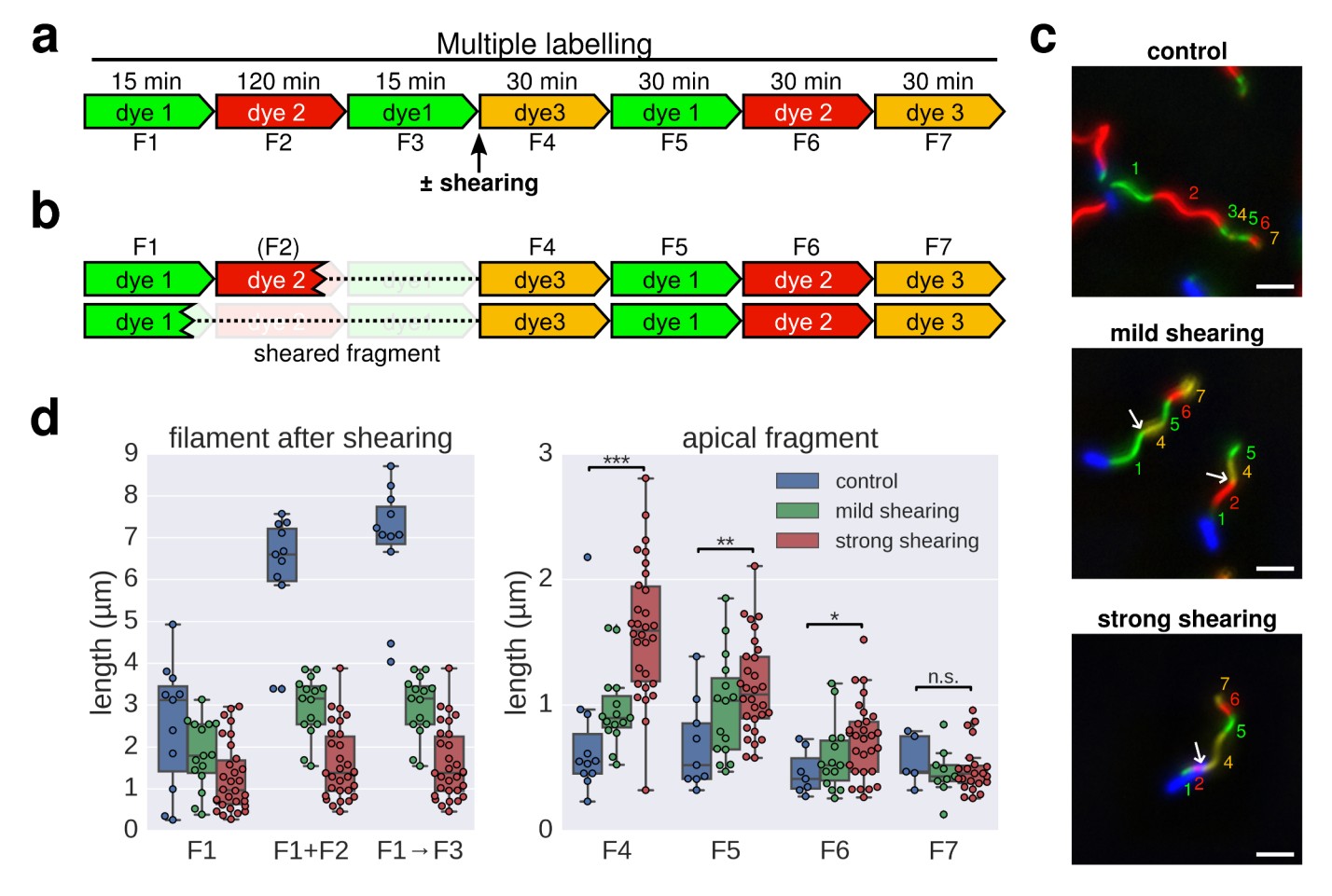

**Figure 4.** Elongation rate of re-growing filaments increases after mechanical shearing. (a) Experimental design to determine filament elongation rate after mechanical shearing using multicolour labelling. (b) A successful shearing event removed fragment F3 and partially or completely fragment F2. (c) Representative example images of control filaments and filaments broken using mechanical shearing forces. Flagellar filaments were sheared by passing the bacterial culture five times (mild shearing) or up to 30 times (strong shearing) in and out of a 22-gauge needle. Scale bar 2 μm. (d) Left panel: length of the basal, cell-attached filament after mechanical shearing demonstrating successful filament breakage. Right panel: length of apical, re-growing filament fragments demonstrating a length-dependent increase in filament elongation rate. The box plots reports the median, the 25th and 75th quartiles and the 1.5 interquartile range. Data points represent individual filament fragments. Statistical significance according to a two-tailed Student's t-test is indicated. F4 strong *vs.* control: p-value=0.000026 (***); F5 strong *vs.* control: p-value=0.002452 (**); F6 strong *vs.* control: p-value=0.034514 (*); F7 strong *vs.* control: not significant (n.s.).

The following figure supplement is available for figure 4:

**Figure supplement 1.** Basal/apical coordinates of sheared and control filaments showing the dispersion of the filament growth data.

To test the requirement of subunit chain formation for flagellum growth in more detail, we generated flagellin mutants truncated for the N- and C-termini that render head-to-tail linkage impossible (*Figure 5c*). All flagellin truncation mutants were secreted, but were deficient in flagellum assembly due to deletions in the D0 and D1 domains needed for filament polymerization and FliS chaperone binding (*Yonekura et al., 2003*) (*Figure 5—figure supplement 1a*, *Figure 5—figure supplement 3*). We expressed those non-chaining, but secreted flagellin mutants in trans to disrupt formation of a chain of wild-type flagellin molecules (*Figure 5d*). Competitive secretion of the flagellin truncation mutants did not affect endogenous flagellin transport during filament formation (*Figure 5c*). Filament extension kinetics were determined using multiple labelling of individual flagellar filaments

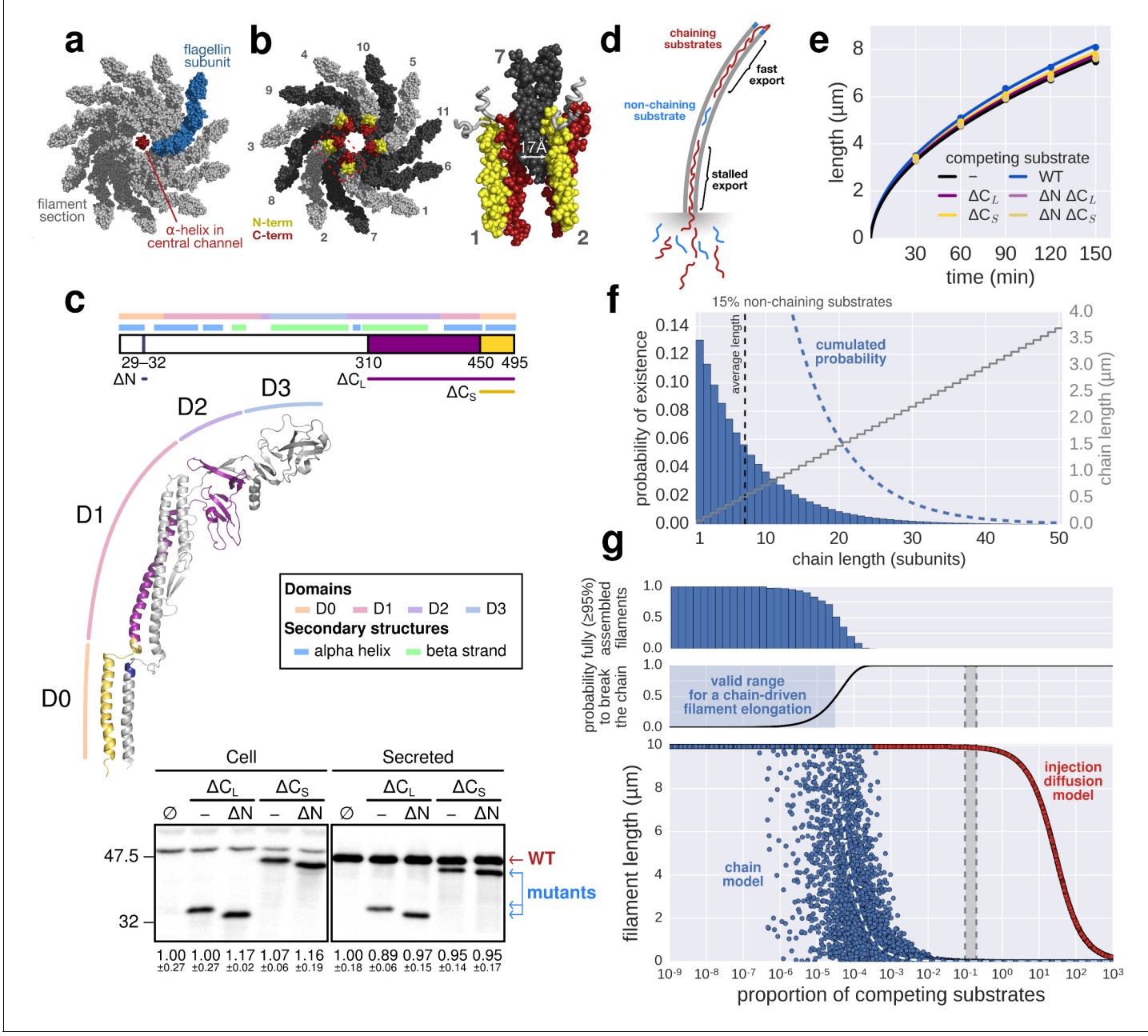

**Figure 5.** The contribution of inter-subunit chains for filament elongation rate. (**a**) The 2 nm wide filament channel only accommodates one folded α-helix. (**b**) The N- and C-termini of successive flagellin molecules are anti-parallel and far apart in the polymerized filament structure. (**c**) Top: Structure, domains, and secondary structures of flagellin FliC (PDB: 1UCU). Mutant flagellins lacking combinations of the N- and C-terminal domains required for head-to-tail coiled-coil chaining (ΔN, ΔC_S, ΔC_L) were co-expressed together with endogenous flagellin (FliC) to disrupt chain formation. Bottom: Flagellin immunoblotting on cellular and secreted fractions (relative full-length flagellin levels report mean ± s.d., n = 3). (**d**) Simultaneous secretion of non-chaining substrates breaks a filament-spanning chain of flagellin molecules. A strict chain model requires the chain to span the entire filament and does not allow for disruptions of the chain. A chain model with the possibility of recovery by diffusion of broken chains is discussed in *Figure 5—figure supplement 1*. (**e**) In situ, multicolour labelling of flagellar filaments during competitive co-expression of chain-disrupting mutant flagellins. The average growth of fits computed from basal/apical coordinates presented in *Figure 5—figure supplement 3c* and as described in *Figure 3c* is shown as a function of time. Basal/apical coordinates were derived from multiple labelling data of individual filaments: *n* = 399 from 89 filaments (−), *n* = 271 from 58 filaments (WT), *n* = 278 from 62 filaments (ΔC_L), *n* = 412 from 93 filaments (ΔN ΔC_L), *n* = 209 from 46 filaments (ΔC_S), *n* = 312 from 73 filaments (ΔN ΔC_S). The fits represent the injection-diffusion model and parameters $k_{on}$ and *D* are given in *Figure 5—source data 1*. (**f**) Probability of existence of *n*-long chains defined by the binomial law. Long chains are highly improbable for a 15% proportion of competing substrates (*i.e.* formation of a more than 2.4 μm long chain (*n* > 33) has a probability of 1%). The bars show the individual probability of existence, the dotted blue line the cumulated

*Figure 5 continued on next page*

*Figure 5 continued*

probability of existence of chains longer than the number on the x-axis. The grey curve indicates the chain length in μm, which reflects that filaments cannot grow longer than a few hundred nanometres with a chain-based mechanism. (g) Simulation of filament growth dependent on inter-subunit chains or the injection-diffusion model in presence of random proportion of competing substrate. The injection-diffusion model fit represents the mean of the multi-labelling data set of *Figure 3* with parameters $k_{on} \approx 27.25$ s$^{-1}$ and $D \approx 5.46 \times 10^{-13}$ m$^2 \cdot$ s$^{-1}$. Dashed white line: median length of the filament for chain-model dependent growth. Grey box: expression range of chain-disrupting mutant flagellins used in panel e and *Figure 5—figure supplement 1a*.

The following source data and figure supplements are available for figure 5:

**Source data 1.** Parameters $k_{on}$ and $D$ of the injection-diffusion model fits of *Figure 5—figure supplement 3*.

**Figure supplement 1.** Filament growth dynamics in the presence of competing non-chaining substrate.

**Figure supplement 2.** Schematic illustration of the chain-model dependent simulation of filament growth.

**Figure supplement 3.** Characterization of chain-disrupting flagellin truncation mutants.

and, similarly, no significant difference was observed when chain-disrupting flagellin mutants were co-expressed (*Figure 5e*, *Figure 5—figure supplement 3c*).

Mathematical modelling of the chain model-dependent filament elongation dynamics predicted a linear growth up to a very long flagellum (>0.1 mm), which is in clear contradiction with the experimental observations (Appendix 2).

## Inhibition of the pmf-dependent protein export prevents filament elongation

Our high-resolution filament growth rate data and the previous observations by *Stern and Berg (2013)* suggested that two major components drive flagellin export: pmf-dependent injection of subunits by the type III export apparatus at the base of the flagellum and diffusion of subunits along the length of the flagellum. We used carbonyl cyanide *m*-chlorophenyl hydrazone (CCCP) to disrupt the pmf, which is required for substrate translocation via the export apparatus into the central channel of the growing flagellar structure (*Minamino and Namba, 2008*; *Paul et al., 2008*). The injection-diffusion model predicts that a decrease in the injection rate $K_{on}$ results in slow, quasi-linear growth for sufficiently small $K_{on}$. As expected, CCCP treatment resulted in impaired filament extension in a dose-dependent manner, which recovered upon removal of the uncoupler (*Figure 6a*, *Figure 6—figure supplement 1*). We hypothesized that in presence of high concentration of CCCP, the injection of substrate would be strongly reduced and result in low-speed growth. As shown in *Figure 6c*, the filament elongation rate for the highest CCCP concentration (~18 nm•min$^{-1}$) was virtually independent of the length of the filament as predicted by the model. Interestingly, some filaments were unaffected by the CCCP treatment, likely due to the action of multidrug transporters (*Lomovskaya and Lewis, 1992*), and displayed kinetics similar to the untreated population (*Figure 6—figure supplement 1d*), highlighting the major contribution of the pmf in energizing export.

## Conclusion

The bacterial flagellum is a remarkably complex nanomachine. Here, we present the first real-time visualization and experimentally supported biophysical model of the dynamic self-assembly process of this large, widely conserved nanomachine. We propose that bacterial flagella grow through an injection-diffusion mechanism (*Figure 6d*), which provides a simple explanation why the flagellar filament does not grow infinitely in the absence of any other length-control mechanism. It appears likely that similar biophysical principles are conserved for effector protein secretion in the evolutionary related, virulence-associated injectisome with important implications for the rational design of novel anti-infectives targeted against type III secretion systems.

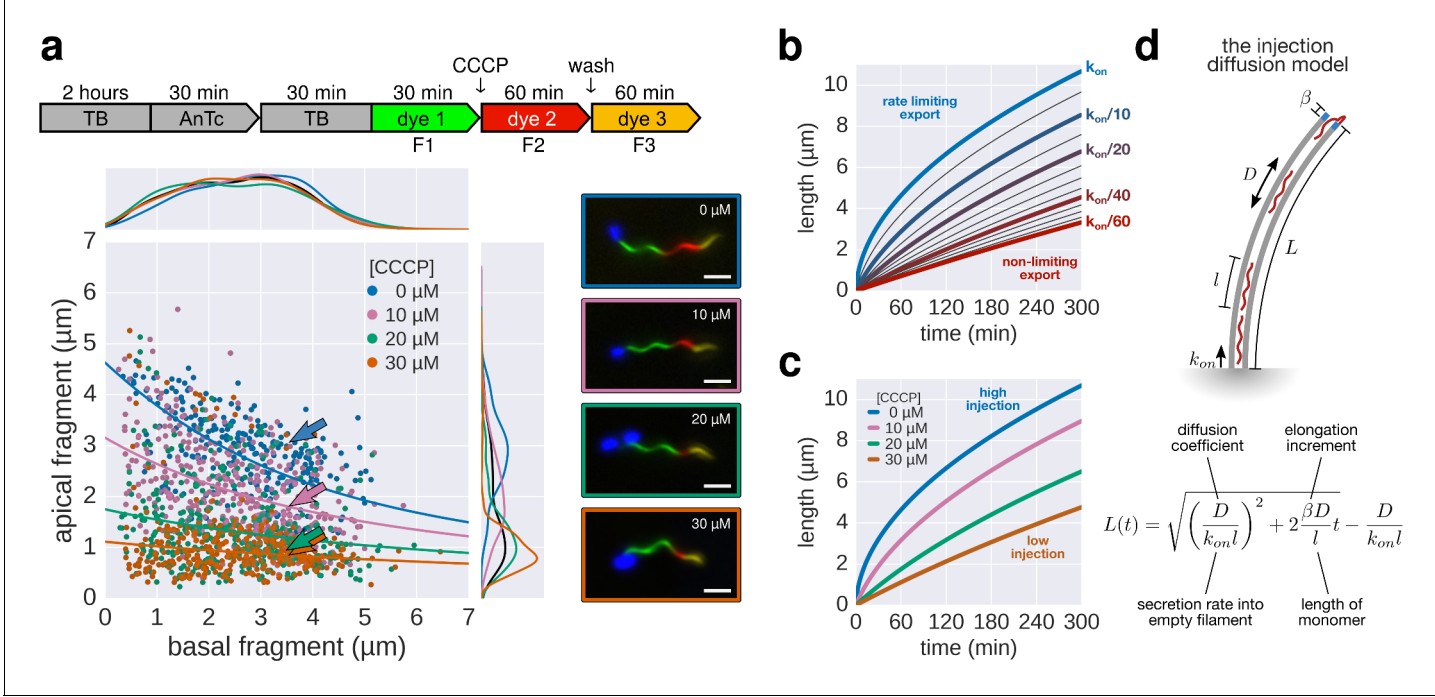

**Figure 6.** The effect of pmf on flagellin injection and filament growth rate. (a) Top: Experimental design. Carbonyl cyanide *m*-chlorophenyl hydrazone (CCCP) reduces the proton motive force (pmf) and was present during growth of the second fragment (60 min) and removed during growth of the third fragment, which allowed the pmf to regenerate. TB: tryptone broth without dye, AnTc: anhydrotetracyline induction of flagella genes. Bottom: Fragment lengths represented as basal/apical (F1/F2) coordinates ($n = 255$ for 0 µM CCCP, $n = 395$ for 10 µM CCCP, $n = 371$ for 20 µM CCCP, $n = 353$ for 30 µM CCCP). The fits represent the injection-diffusion model with parameters $D \approx 5.25 \times 10^{-13}$ m$^2 \cdot$ s$^{-1}$, and $k_{on} \approx 26.10, 3.19, 1.19, 0.70$ s$^{-1}$ for 0 µM, 10, 20, 30 µM CCCP respectively. Representative fluorescent microscopy images of labelled flagella and matching coordinates are highlighted by coloured frames and arrows. Scale bar 2 µm. (b) Filament length as a function of time for decreasing values of $k_{on}$. For small values of $k_{on}$, the injection rate but not flagellin transport is rate-limiting, which results in a quasi-linear growth. (c) Growth curves for the CCCP raw data of panel a determined by fitting the data to the injection-diffusion model with a fixed parameter a. The values for $k_{on}$ decrease by a factor of 8 (10 µM CCCP), 22 (20 µM CCCP), and 38 (30 µM CCCP), compared to the untreated control. (d) Equation and biophysical parameters of the injection-diffusion model.

The following figure supplement is available for figure 6:

**Figure supplement 1.** Supporting data for effect of CCCP inhibition of the pmf-dependent protein export on flagella growth rate.

## Materials and methods

### Bacteria, plasmids and media

*Salmonella enterica* serovar Typhimurium strains and plasmids used in this study are listed in *Table 1*. Lysogeny broth (LB) contained 10 g of Bacto-Tryptone (Difco), 5 g of yeast extract, 5 g of NaCl and 0.2 ml of 5N NaOH per litre. Soft agar plates used for motility assays contained 10 g of Bacto-Tryptone, 5 g of NaCl, 3.5 g of Bacto-Agar (Difco) and 0.2 ml of 5N NaOH per liter. Tryptone broth (TB) contained 10 g of Bacto-Tryptone and 5 g of NaCl. Ampicillin was added to the medium at a final concentration of 100 µg/ml, L-arabinose at a final concentration of 0.2% and anhydrotetracyline at a final concentration of 100 ng/ml if required.

### DNA manipulations

DNA manipulations were carried out as described before (*Hara et al., 2011*). Site-directed mutagenesis was carried out using QuickChange site-directed mutagenesis method as described by Agilent Technologies, Santa Clara, CA, USA. DNA sequencing reactions were carried out using BigDye v3.1 as described in the manufacturer's instructions (Applied Biosystems, Foster City, CA, USA), and then the reaction mixtures were analysed by a 3130 Genetic Analyzer (Applied Biosystems).

**Table 1.** Strains and plasmids used in this study.

| Strain | Relevant characteristics | Source or reference |
|---|---|---|
| SJW1103 | *Salmonella enterica* serovar Typhimurium wild-type strain SJW1103 for motility and chemotaxis | (*Yamaguchi et al., 1984*) |
| TM113 | SJW1103 Δ*fliC* | T. Miyata, unpublished |
| NH001 | SJW1103 Δ*flhA* | (*Hara et al., 2011*) |
| MM1103iS | SJW1103 Δ*fliS::km* | (*Furukawa et al., 2016*) |
| MM1103gK | SJW1103 *flgK::Tn10* | This study |
| MM1103gKiS | SJW1103 Δ*fliS::km flgK::Tn10* | This study |
| MM1103CPOP | SJW1103 ΔP$_{fliC}$::*tetRA*-62 | This study |
| TH437 | *Salmonella enterica* serovar Typhimurium wild-type strain LT2 | lab collection |
| TH15801 | LT2 P$_{flhDC}$5451::Tn*10d*Tc[del-25] Δ*hin*-5717::FCF | lab collection |
| EM1237 | LT2 Δ*araBAD*1026::*fliC* Δ*fliC*7861::FRT Δ*hin*-5717::FCF P$_{flhDC}$5451::Tn*10d*Tc[del-25] | This study |
| EM2046 | LT2 Δ*hin*-5717::FRT *fliC*6500 (T237C) P$_{flhDC}$5451::Tn*10d*Tc[del-25] | This study |
| EM2400 | LT2 Δ*hin*-5717::FRT *fliC*6500(T237C) Δ*araBAD*1005::FRT P$_{flhDC}$5451::Tn*10d*Tc[del-25] | This study |
| EM4076 | LT2 Δ*hin*-5717::FRT *fliC*7746::3xHA (Δaa201-213::3xHA) *motA*5461::MudJ P$_{flhDC}$5451::Tn*10d*Tc[del-25] Δ*sseA-ssaU*::FCF (deletes Spi-2) | This study |
| Plasmids | Relevant characteristics | Source or reference |
| pBAD24 | Expression vector | Invitrogen |
| pAOA001 | pBAD24/FliC | This study |
| pAOA002 | pBAD24/FliC(Δ29–32) | This study |
| pAOA003 | pBAD24/FliC(Δ11–18) | This study |
| pAOA004 | pBAD24/FliC(Δ11–18/Δ29–32) | This study |
| pAOA005 | pBAD24/FliC(Δ310–495) | This study |
| pAOA006 | pBAD24/FliC(Δ29–32/Δ310–495) | This study |
| pAOA007 | pBAD24/FliC(Δ450–495) | This study |
| pAOA008 | pBAD24/FliC(Δ29–32/Δ450–495) | This study |

## Motility assays in soft agar

To check motility of the *Salmonella* SJW1103 (wild-type) and TM113 (Δ*fliC*) cells carrying a pBAD24-based plasmid encoding wild-type or FliC deletion variants, motility assays were performed in soft agar plates. Single colonies of the cells were inoculated into soft agar plates containing ampicillin and 0.2% arabinose. Plates were then incubated at 30°C for the required period of time. Their motility was observed as a ring of migrating cells emanating from the point of inoculation.

## Flagellin transport assay

*Salmonella* cells were grown with shaking in 5 ml of LB at 30°C until the cell density had reached an OD$_{600nm}$ of approximately 1.0–1.2. To see the effect of the flagellar filament on flagellin transport during filament assembly, the cultures were heated at 65°C for 5 min to depolymerize the filaments into flagellin monomers and were centrifuged to obtain cell pellets and culture supernatants, which contains the cytoplasmic flagellin subunits and flagellins transported by the flagellar type III export apparatus, respectively. To test the effect of flagellin subunit linkage on the flagellar growth rate (compare *Figure 5c*), strain MM1103CPOP carrying a pBAD24-based plasmid encoding FliC(Δ310–495), FliC(Δ29–32/Δ310–495), FliC(Δ450–495) or FliC(Δ29–32/Δ450–495) was grown with shaking in 5 ml of LB containing ampicillin at 30°C until the cell density had reached an OD$_{600}$ of approximately 0.6–0.8. To induce the expression of chromosomally encoded wild-type FliC (from a tetracycline-inducible promoter in the native *fliC* locus) and its deletion variant (from an arabinose-inducible

promoter encoded on pBAD24), we added tetracycline and L-arabinose at the final concentrations of 15 µg/ml and 0.2%, respectively, and the incubation was continued for another hour. The cultures were directly heated at 65°C for 5 min, followed by centrifugation to obtain cell pellets and culture supernatants. Cell pellets were resuspended in the SDS-loading buffer, normalized to a cell density to give a constant amount of cells. Proteins in the culture supernatants were precipitated by 10% trichloroacetic acid, suspended in the Tris/SDS loading buffer and heated at 95°C for 3 min. After SDS-PAGE, both CBB-staining and immunoblotting with polyclonal anti-FliC antibodies were carried out as described before (*Minamino and Macnab, 1999*). Detection was performed with an ECL plus immunoblotting detection kit (GE Healthcare, Tampa, FL, USA). At least six independent experiments were performed.

## Flagellin leakage measurements during filament assembly

*Salmonella* cells were grown with gentle shaking in 5 ml of LB at 30°C until the cell density had reached an $OD_{600}$ of approximately 1.0. After centrifugation, the cell pellets and the culture supernatants were collected separately. The culture supernatants were ultracentrifuged at 85,000 $\times$ *g* for 1 hr at 4°C and the pellets and the supernatants, which contain flagellar filaments detached from the cell bodies during shaking culture and flagellin monomers leaked out the culture media during filament formation, respectively, were collected separately. The cell pellets were suspended in 5 ml PBS and then were heated at 65°C for 5 min, followed by centrifugation to obtain the cell pellets and supernatants, which contained the cytoplasmic flagellin molecules and depolymerized flagellin monomers, respectively. The cell pellets and the pellet fractions after ultracentrifugation were resuspended in the SDS-loading buffer, normalized to the cell density to give a constant amount of cells. Proteins in the supernatants were precipitated by 10% trichloroacetic acid, suspended in Tris/SDS loading buffer and heated at 95°C for 3 min. After SDS-PAGE, both CBB-stating and immunoblotting with polyclonal anti-FliC antibodies were carried out. At least six independent experiments were performed.

## Electron microscopy observation of negatively stained *Salmonella* cells

*Salmonella* cells were exponentially grown with gentle shaking in 5 ml LB at 30°C. 5 µl of the cell culture were applied to carbon-coated copper grids and negatively stained with 0.5% (W/V) phosphotungstic acid. Micrographs were recorded at a magnification of 1200$\times$ with a JEM-1010 transmission electron microscope (JEOL, Tokyo, Japan) operating at 100 kV.

## Microscopy of flagellar filaments

For immunolabelling of flagellar filaments, polyclonal anti-FliC and anti-rabbit IgG antibodies conjugated with Alexa Fluor 488 and 594 (Invitrogen, Carlsbad, CA, USA) were used as described (*Erhardt et al., 2011*; *Minamino et al., 2014*).

For in situ labelling of flagellar filaments of the $FliC^{T237C}$ cysteine replacement mutant, an overnight culture was diluted 1:100 into 10 ml fresh TB in a 125 ml flask and grown at 30°C for 2 hr until $OD_{600nm}$ of 0.6. Production of the flagellar master regulatory operon *flhDC* was induced by addition of 100 ng/ml anhydrotetracycline (AnTc) for 30 min. Afterwards, the culture was centrifuged for 3 min at 2500 $\times$ *g*, resuspended in 10 ml fresh TB and grown at 30°C for 30 min. An aliquot was transferred to a 2 ml Eppendorf tube and grown with shaking at 30°C for 30 min in the presence of 10–25 µM Alexa or DyLight-coupled maleimide dye (ThermoFisher, Tampa, FL, USA). After the incubation, the dye was removed by centrifugation for 2 min at 2500 $\times$ *g*. The culture was resuspended in 1 mL fresh TB and incubated for additional 30 min in the presence of 10–25 µM Alexa or DyLight-coupled maleimide dye at 30°C. Dye removal and incubation with DyLight-coupled maleimide dye was repeated to label up to six flagellar filament fragments. After the final labelling period, the bacteria were resuspended in PBS and an aliquot was applied to a custom-made flow cell (*Wozniak et al., 2010*) with the modification of using Polysine microscope slides (ThermoFisher). Non-adhering cells were flushed by addition of PBS and bacteria were fixed by addition of 2% formaldehyde, 0.2% glutaraldehyde in PBS for 5 min, followed by a washing step with PBS. Fluoroshield mounting medium (Sigma-Aldrich, St. Louis, MO, USA) was added and the cells were observed by fluorescent microscopy using a Zeiss (Oberkochen, Germany) Axio Observer microscope at 100$\times$ magnification. Fluorescence images were analysed using ImageJ software version 1.48 (National Institutes of Health).

Continuous flow in situ immunostaining of 3× hemagglutinin epitope tagged FliC filaments was performed as described by *Berk et al. (2012)* with the following adaptions. Strain EM4076 expressing mCherry from pZS*12-mCherry (mCherry under control of P*lac* [*Lutz and Bujard, 1997*]) was grown to mid-log phase in M9-glucose minimal medium supplemented with 0.2% casamino acids and 0.1% bovine serum albumin (BSA) and induced for 30 min with 100 ng/ml AnTc. Bacteria were diluted 10-fold, and applied to a continuous flow CellASIC microfluidic plate (B04A; Merk Millipore, Billerica, MA, USA). Approximately 10 nM anti-HA Alexa Fluor488 fluorochrome-coupled primary antibodies (Thermo Fisher A-21287, final concentration 1 µg/ml) were added to the flow medium, which was identical to the above mentioned growth medium without addition of AnTc. Cells were imaged at 30°C with a temperature-controlled Olympus total internal reflection fluorescence microscope equipped with a water-cooled Hamamatsu (Hamamatsu City, Japan) ImageEM C9100-13 with a pixel size of 160 µm using a NA1.4 100× objective and an additional 1.6× tubular lens at a highly-inclined above-critical angle. To image anti-HA Alexa Fluor488 decorated flagellin and mCherry, a 488 nm diode laser set to 0.25 mW and a 561 nm solid-state laser set to 0.85 mW were used. Images were taken every 10 s with exposure times of 15 msec for 488 nm and 8 msec for 561 nm at low camera gain settings.

## Data reporting

No statistical methods were used to predetermine sample size. The experiments were not randomized and the investigators were not blinded to allocation during experiments and outcome assessment.

## Statistical analysis

Biochemistry experiments were performed at least three times and representative experiments are reported in the figures. Where indicated, mean values and standard deviations were obtained from at least three independent biological replicates. All microscopy experiments were performed at least twice and the figures present individual data points of a representative experiment. Box plots report the median (in red), the 25th and 75th quartiles and the 1.5 interquartile range. Error bars of bar graphs represent the 95% confidence interval of mean estimation.

## Fitting experimental data by the growth model

To compare the model with data, we need to find a best fit for the parameters $a$ and $b$ using the growth function (*Equation 4*). Accordingly, note that if $F_1$ is the amount of filament growth in time $\Delta T$ following an initial growth of length $F_0$, then

$$\int_{F_0}^{F_0+F_1} (b+L)dL = a\Delta T, \tag{7}$$

which reduces to the equation

$$L(L+2b)\Big|_{F_0}^{F_0+F_1} = 2a\Delta T, \tag{8}$$

or

$$bF_1 + \frac{1}{2}\left(2F_0F_1 + F_1^2\right) = a\Delta T. \tag{9}$$

This could be solved for $F_0$ as a function of $F_1$ and then fitted by standard regression to find parameters $a$ and $b$. However, to do so would ignore the important fact that there is measurement error in both of the measurements $F_0$ and $F_1$. Consequently, a different method of fitting this curve is needed.

The method used here is to seek numbers $W_0$ and $W_1$, which are approximations to $F_0$ and $F_1$ and satisfy the relationship

$$bW_1 + \frac{1}{2}\left(2W_0W_1 + W_1^2\right) = a\Delta T \tag{10}$$

This can be done by minimizing the function

$$E = \sum^{N}\left(\left(F_0 - W_0\right)^2 + \left(F_1 - W_1\right)^2 + \lambda\left(bW_1 + \frac{1}{2}\left(2W_0W_1 + W_1^2\right) - a\Delta T\right)^2\right), \tag{11}$$

where $\lambda$ is a fixed constant. In this way, both $F_0$ and $F_1$ are treated as noisy data values which need to be fitted.

However, for this analysis, we found it better to introduce the change of variables $L = b\frac{U}{1-U} = g(U)$, $U = \frac{L}{L+b}$ and then to find numbers $U_0$ and $U_1$, $\alpha = \frac{a\Delta T}{b^2}$ and $b$ so that

$$E = \sum^{N}\left(\left(F_0 - bg\left(U_0\right)\right)^2 + \left(F_0 + F_1 - bg\left(U_1\right)\right)^2 + \lambda\left(f\left(U_1\right) - f\left(U_0\right) - \alpha\right)^2\right) \tag{12}$$

is minimized, where $f(U) = \frac{1}{b^2}\left(bL + \frac{1}{2}L^2\right) \equiv \frac{1}{2}\frac{U\left(2-U\right)}{\left(1-U\right)^2}$. The minimization of $E$ is equivalent to finding the solution of the system of $2N + 2$ nonlinear algebraic equations

$$\frac{\partial}{\partial\alpha} : \sum^{N}\left(f\left(U_1\right) - f\left(U_0\right) - \alpha\right) = 0, \tag{13}$$

$$\frac{\partial}{\partial b} : \sum^{N}\left(F_0 - bg\left(U_0\right)\right)g\left(U_0\right) + \sum^{N}\left(F_0 + F_1 - bg\left(U_1\right)\right)g\left(U_1\right) = 0, \tag{14}$$

$$\frac{\partial}{\partial U_0} : b\left(F_0 - bg\left(U_0\right)\right)g'\left(U_0\right) + \lambda\left(fU_1\right) - f\left(U_0\right) - \alpha\left(f'\left(U_0\right)\right) = 0, \tag{15}$$

$$\frac{\partial}{\partial U_1} : -b\left(F_0 + F_1 - bg\left(U_1\right)\right)g'\left(U_1\right) + \lambda\left(f\left(U_1\right) - f\left(U_0\right) - \alpha\right)f'\left(U_1\right) = 0. \tag{16}$$

This system of equations is readily solved with an iterative solution method such as Newton's Method, details of which are not described here.

Once $U_0$ and $U_1$ are known, so also are $W_0 = b\frac{U_0}{1-U_0}$ and $W_1 = b\frac{U_1}{1-U_1} - W_0$. From this we can estimate the time at which the $F_0$ phase of growth ended to be

$$t_0 = \frac{1}{a}\left(bW_0 + \frac{W_0^2}{2}\right), \tag{17}$$

and the time at which the $F_1$ growth phase ended is $t_1 = t_0 + \Delta T$. This information enables us to plot the growth curve and plot the $F_0$ and $F_1$ measurements at the appropriate times.

## Simulation of filament growth in presence of competing substrates

Substrates arriving at the export gate were randomly chosen with a probability $p = r / (1+r)$ to be a competing substrate (i.e., non-chaining or not incorporated in the filament), where $r$ is the ratio of competing molecules relative to flagellin. The following rules were used:

1. 1)Unaltered growth (i.e. without competing substrate) followed the kinetics determined experimentally: $L(t) = -b + \sqrt{b^2 + 2at}$ with $a = 0.208$ and $b = 0.271$. Note that this observed $L(t)$ dependence does not preclude the chain formation model, which may also lead to the same form of growth kinetics, but only for growth without any chain breakage. However, the general conclusion is independent of the specific growth form, e.g., a linear growth kinetics lead to the same conclusion.
2. 2)Competition for injection was considered to induce, per competing substrate, a delay, with a time scale that is taken to be the same as the injection time $t_{on} = k_{on}^{-1} \cong 38\,ms$.
3. 3)For the chain model, chain breakage due to competing substrates induced either an arrest of elongation in the strict chain model (see *Figure 5*) or a delay in growth in the chain recovery model (see *Figure 5—figure supplement 1* ) with a delay time given by $(t_b - t_a)$, where:

i. (i)$t_b$ is the time required for the basal chain to diffuse to the tip defined as $t_b = \frac{\eta L_a^2}{D_0}$ with $D_0$ the diffusion coefficient of a flagellin monomer, $L_a$ the length to diffuse to reach the tip, and $n$ the number of subunits in the chain.

ii. $t_a$ is the time required for the apical chain to fold according to the kinetic of unaltered growth.

In *Figure 5* panel g and *Figure 5—figure supplement 1* panel d, we simulated filament growth over 250 min for 20,000 filaments and assumed a mechanism based on the chain model (strict in blue, with recovery in yellow) or the injection-diffusion model (in red), in the presence of a random proportion of competing substrates ($r$) between $10^{-9}$ and 1,000. The simulation of chain model-dependent filament growth is illustrated in *Figure 5—figure supplement 2*.

The range of competing substrates compatible with a chain-driven elongation is very low ($<10^{-4}$–$10^{-5}$), while the injection-diffusion model allows for robust filament growth over a much broader range of competing substrate (up to about a 10-fold excess of competing substrates).

Complementary to the simulation, the median length of the filament under chain model-dependent growth and in presence of competing substrates can be calculated as follows:

The probability of sequentially forming a chain of exact length $n$ is $P_n = p^n(1 - p)$.

The expected number of molecules in the chain is:

$$E(p) = (1 - p) \sum_n n p^n = \frac{p}{1 - p} = \frac{1}{x} \tag{18}$$

Thus, the median length of a filament grown from a continuous chain is $k\beta$, where $\beta = 0.47$ nm and $k$ can be determined by:

$$\frac{1}{2} = \sum_n^k P_n = \sum_n^k (1 - p)p^n = 1 - p^{k+1}, \tag{19}$$

which leads to:

$$k = \frac{\ln 2}{\ln(1 + x)} - 1. \tag{20}$$

## Acknowledgements

We thank to Howard Berg, David F Blair and Kelly T Hughes for useful discussions and generous donation of strains, Takuma Fukumura and Nadine Körner for expert technical assistance and Tomoko Miyata for her kind gift of TM113. This work was supported in part by JSPS KAKENHI grant 25000013 (to KN) and 26293097 (to TM) and MEXT KAKENHI grant 24117004 and 15H01640 (to TM), the CFI, NSERC and CREATE (to SR), the Max Planck Society (to EC), NIH grant R01GM081747 (to. YT) and the Helmholtz Association Young Investigator grant VH-NG-932 and the People Programme (Marie Curie Actions) of the Europeans Unions' Seventh Framework Programme grant 334030 (to ME). TTR gratefully acknowledges fellowship support by the Alexander von Humboldt Foundation.

Primary correspondence and requests for materials should be addressed to M.E. (marc.erhardt@helmholtz-hzi.de).

Correspondence concerning the mathematical injection-diffusion model should be addressed to J. P.K. (keener@math.utah.edu) and Y .T . (yuhai@us.ibm.com). Correspondence concerning flagellin competition experiments should be addressed to T.M. (tohru@fbs.osaka-u.ac.jp) and K.N. (keiichi@fbs.osaka-u.ac.jp).

## Additional information

### Funding

| Funder | Grant reference number | Author |
|---|---|---|
| Helmholtz-Gemeinschaft | VH-NG-932 | Marc Erhardt |
| Max-Planck-Gesellschaft | | Emmanuelle Charpentier |

| National Institutes of Health | R01GM081747 | Yuhai Tu |
|---|---|---|
| European Commission | 334030 | Marc Erhardt |
| Japan Society for the Promotion of Science | 25000013 | Keiichi Namba |
| Natural Sciences and Engineering Research Council of Canada | | Simon Rainville |
| Alexander von Humboldt-Stiftung | | Thibaud T Renault |
| Japan Society for the Promotion of Science | 26293097 | Tohru Minamino |
| Ministry of Education, Culture, Sports, Science and Technology | 24117004 | Tohru Minamino |
| Ministry of Education, Culture, Sports, Science and Technology | 15H01640 | Tohru Minamino |

The funders had no role in study design, data collection and interpretation, or the decision to submit the work for publication.

### Author contributions

TTR, Conceptualization, Formal analysis, Investigation, Visualization, Writing—original draft, Writing—review and editing; AOA, Formal analysis, Investigation, Writing—review and editing; TB, Investigation, Writing—review and editing; GP, SR, Conceptualization, Writing—review and editing; EC, CCG, Funding acquisition, Writing—review and editing; YT, JPK, Conceptualization, Formal analysis, Writing—original draft, Writing—review and editing; KN, Conceptualization, Supervision, Funding acquisition, Writing—review and editing; TM, ME, Conceptualization, Formal analysis, Supervision, Funding acquisition, Investigation, Writing—original draft, Writing—review and editing

### Author ORCIDs

Thibaud T Renault, http://orcid.org/0000-0002-1530-2613
Anthony O Abraham, http://orcid.org/0000-0002-8710-1351
Călin C Guet, http://orcid.org/0000-0001-6220-2052
Marc Erhardt, http://orcid.org/0000-0001-6292-619X

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

## Differences in the experimental design explain the apparent differences in filament growth rate observed by *Turner et al. (2012)*

In the following, we describe differences in the experimental setup of *Turner et al. (2012)* from ours, which readily explain the apparent differences in growth for short filaments and thus reconcile our results with the previous work.

First, the filament labelling protocol used by *Turner et al. (2012)* included multiple, long-term centrifugation steps to remove excess dye and wash the samples. We found the maleimide labelling of the exposed cysteine residue to be very specific and a single centrifugation step was sufficient to remove residual maleimide dye prior to the next labelling step. Thus, our samples were exposed to only 3–6 short-term, low speed centrifugation steps for a total duration of 6–12 min (for the triple and multi-colour labelling, respectively). In comparison, the samples of *Turner et al. (2012)* were exposed to three long-term, repeated centrifugation and resuspension washes for a total duration of 187 min. Flagellar filaments can easily break due to the shearing forces of centrifugation or repeated pipetting. It thus appears possible that the frequent centrifugation steps increased the frequency of broken filaments during the experiment of *Turner et al. (2012)*, which possibly contributed to the fraction of first-fragment-only (= broken or stalled) filaments observed in *Figure 3* of *Turner et al. (2012)*. While we do not know how to exactly convert the experimental conditions of *Turner et al. (2012)* quantitatively into our model, we performed a simulation of increasing fractions of broken filaments using our multi-color data set. As shown in *Figure 3—figure supplement 2*, the broken/stalled filaments accumulate on the x-axis and below the fit curve and thus result in a quasi-linear fit of the complete filament growth data set.

Further, the filament labelling protocol of *Turner et al. (2012)* included a long-term incubation step overnight at 7°C in contrast to the in situ labelling in our experimental setup, where we added the maleimide dyes during normal culture growth to minimize perturbations. It appears possible that long-term incubation of samples at low temperatures might have negatively affected the injection rate of flagellin subunits into the growing filament *e.g.* by alterations in protein translation efficiency or changes in the available proton motive force. Supporting this possibility, we also observed quasi-linear growth of the filament if the injection rate and not flagellin transport was rate-limiting (*Figure 3—figure supplement 2*, *Figure 6*).

Finally, the growth rate data of *Turner et al. (2012)* contain only few measurements of short filaments for which we observed faster growth (the majority of measured filaments were 4–6 µm long), while our filament growth data report filaments ranging from 0.2–10 µm. In summary, the extended range of filament lengths, the possibility of broken/stalled filaments and possible perturbations of the injection rate reconcile our data with the reported filament growth data of *Turner et al. (2012)* and explains why we observed a clear length-dependent decrease in growth rate.

## Appendix 2

# Dynamics of the chain model and comparison to the injection-diffusion model

In the following, we describe the differences between the chain model proposed by *Evans et al. (2013)* and our injection-diffusion model for flagellum growth. For a chain of length $L$, there are a total of $N = L/l$ monomers, where $l = 74$ nm is the length of the unfolded monomers. For a freshly arrived lead-monomer (of the chain) at the distal end to grow into the flagellum, two things have to happen:

i.  i)The lead-monomer will take some time to initiate the folding and then 'crystallize', which takes a total time $t_f$.
ii.  The whole chain has to move a distance $l$ driven by the folding.

After these two steps, a new lead-monomer will arrive and the chain is ready to go through the same process to continue growth. If the folding of the monomer provides a force $f_0$, and the friction constant of the chain is $N\eta_0$, where $\eta_0$ is the monomer friction constant, then the moving speed is $f_0 / (N\eta_0)$, and the moving time is:

$$t_m = l/(f_o/N\eta_0)) = \frac{Nl\eta_0}{f_0} = \frac{\eta_0 L}{f_0}. \tag{A1}$$

Now, if we assume $t_f$ is mostly dominated by the initiation time, the total time it takes for one monomer to be fully incorporated into the flagellum and to be ready again for further growth is:

$$t_g = t_f + t_m, \tag{A2}$$

which leads to an equation of growth with the same form to our injection-diffusion growth model:

$$\frac{dL}{dt} = \beta/t_g = \frac{\beta}{t_f + \frac{\eta_0}{f_0}L} \tag{A3}$$

Despite its similar form, the meaning of this equation is completely different from our model. For comparison, our injection-diffusion model is given here again:

$$\frac{dL}{dt} = \frac{\beta}{k_{on}^{-1} + \frac{l}{D_0}L}, \tag{A4}$$

where $D_0$ is the diffusion constant of a monomer.

The two growth models differ in the two terms in the denominators in the right hand side of the growth equations. A careful look at these two terms can help us distinguish these two models as discussed in the following:

The first term in the denominators controls the initial linear growth. In our injection-diffusion model, it is given by the injection time , $t_{on} \equiv k_{on}^{-1}$, which is the time for a monomer to be unfolded and injected at the basal end of the flagellum; whereas in the chain-forming model, it is given by the crystallization/folding time $t_f$ at the distal end. The values of $k_{on}$ or $t_g$ can be obtained from the initial growth rate measured in the experiments.

The second term in the denominator is more revealing. Due to the extremely large folding force $f_0$, the second term (the $L$-dependent term) is much smaller in the chain model than in our model.

Since this second term is responsible for the slower $\sqrt{t}$, the chain-forming model would predict a linear growth dynamics up to a very large flagellum length, beyond the longest flagellum length observed, which is obviously in contradiction with the observations.

Quantitatively, we can compute the ratio of the two terms from the two models by using the Einstein relationship $D_0 \eta_0 = k_B T$:

$$\frac{\left(\frac{\eta_0}{f_0}L\right)}{\left(\frac{l}{D}L\right)} = \frac{k_B T}{f_0 l} = \frac{k_B T}{\Delta E_f}, \tag{A5}$$

where $\Delta E_f = f_0 l$ should be comparable to the folding free energy, which is $\gg k_B T$ with $k_B T$ the thermal energy at room temperature ($k_B T \approx 4$ pN $\cdot$ nm).

The second term of the chain-forming model is negligible with values of $f_0 = 10–30$ pN (**Evans et al., 2013**) and $l = 74$ nm. Accordingly, as noted by **Evans et al. (2013)**, friction is not the rate-limiting factor for movement of an inter-subunit chain in the channel and would follow the rate of crystallization at the filament tip.

In the chain-forming model, the crossover length, $L_c = t_f f_0 / \eta_0$, is much larger (could be > 100 μm) than in our model (~0.25–2 μm), and one would not observe the cross over to the slower $\sqrt{t}$ growth, which is in clear contradiction with the experimental observations.

All the above assumes an end-to-end chain does form, however. Our experimental evidence suggests that such long chains do not form (**Figure 5**); and perhaps even more importantly, the growth works fine with the injection-diffusion model where there is no chain.

