## [Decision Letter]

Thank you for submitting your article "Bacterial flagella grow through an injection-diffusion mechanism" for consideration by *eLife*. Your article has been reviewed by two peer reviewers, and the evaluation has been overseen by Frank Jülicher as the Reviewing Editor and Richard Aldrich as the Senior Editor. The following individuals involved in review of your submission have agreed to reveal their identity: Richard M Berry (Reviewer #1); Victor Sourjik (Reviewer #2).

The reviewers have discussed the reviews with one another and the Reviewing Editor has drafted this decision to help you prepare a revised submission.

Summary:

The manuscript of Renault et al. addresses a highly interesting and controversial topic of how bacterial flagellar filaments, by far the most prominent extracellular structure known in bacteria, are assembled. It is well established that flagellar assembly occurs at the tip, meaning that the subunits need to travel across the pore within a filament is a semi-unfolded state and polymerize at the exit site. However, it remains unclear how this mode of assembly is regulated, and particularly how it leads to the formation of filaments of defined length. In their study, Renault et al. combine elegant microscopy experiments with theory to provide very strong evidence in favour of the injection-diffusion model of assembly. The authors further provide convincing experimental demonstration that mutants lacking chain-forming domains that are implicated in a competing model of flagellar export are secreted without disrupting normal filament growth. This provides strong evidence against the chain model. Altogether, the results of this manuscript are compelling and the analysis is solid. There is only one important point that needs to be addressed prior to publication.

Essential revisions:

The results of this manuscript apparently contradict a previous study (Turner et al., 2012) that concluded that growth rate of flagellar filaments is length-independent. The authors clearly cite that previous study and point to the discrepancy in the Results, and they also briefly mention differences in the experimental design. However, it would be important to discuss this discrepancy more extensively, because it relates to the central conclusion of this manuscript. The authors' analysis presented here is convincing, but it is still unclear why different conditions used by Turner et al. led to the opposite conclusion. Possible reasons for previous reports of slower growth should be pointed out explicitly. Differences with previous work that could be relevant should be explained in the Discussion. Since the authors have a mathematical model at hand, they might even be able to simulate filament growth under conditions used by Turner et al., to reconcile their results with that previous work.

---

## [Author Response]

*Essential revisions:*

*The results of this manuscript apparently contradict a previous study (Turner et al., 2012) that concluded that growth rate of flagellar filaments is length-independent. The authors clearly cite that previous study and point to the discrepancy in the Results, and they also briefly mention differences in the experimental design. However, it would be important to discuss this discrepancy more extensively, because it relates to the central conclusion of this manuscript. The authors' analysis presented here is convincing, but it is still unclear why different conditions used by Turner et al. led to the opposite conclusion. Possible reasons for previous reports of slower growth should be pointed out explicitly. Differences with previous work that could be relevant should be explained in the Discussion. Since the authors have a mathematical model at hand, they might even be able to simulate filament growth under conditions used by Turner et al., to reconcile their results with that previous work.*

This is an important point and we thank the referees for raising it. We are convinced that differences in the experimental setup and data analysis of Turner et al. 2012 readily explain the apparent differences in growth for short filaments and thus reconcile our results with the work of Turner et al. 2012.

In particular, we believe that multiple, long-term centrifugation steps and long-term incubation at low temperatures of the experimental setup of Turner et al. 2012 might have negatively affected the injection rate and frequency of filament breakage. We thus took great care to optimize our experimental setup in order to minimize such perturbations of flagellum growth.

Accordingly, we improved our filament labelling approach to exchange dyes multiple times with minimal perturbation of bacterial growth. Importantly, we label the filaments in situ during normal culture growth, which does not affect the flagellum growth rate (Figure 2—figure supplement 1).

We include now a detailed comparison of our experimental approaches with the approach of Turner et al. 2012 in the revised manuscript (Appendix 1) and the main differences are summarized below:

1) The filament labelling protocol used by Turner et al. 2012 included multiple, long-term centrifugation steps to remove excess dye and wash the samples. Each of those centrifugation (centrifugation 3-4 times at 1.400 × g for 12 minutes) and resuspension manipulations spanned a duration of 51–68 min per labelling in contrast to a single, short 2 minutes centrifugation step at 2.500 × g per labelling in our experimental setup.

We found the maleimide labelling of the exposed cysteine residue to be very specific and a single centrifugation step was sufficient to remove residual maleimide dye prior to the next labelling step. Thus, our samples were exposed to only 3–6 short-term, low speed centrifugation steps for a total duration of 6–12 minutes (for the triple and multi-colour labelling, respectively). In comparison, the samples of Turner et al. 2012 were exposed to three long-term, repeated centrifugation and resuspension washes for a total duration of 187 minutes.

It is important to note that flagellar filaments can easily break due to the shearing forces of centrifugation or repeated pipetting. It thus appears possible that the frequent centrifugation steps increased the frequency of broken filaments during the experiment of Turner et al. 2012, which likely contributed to the large fraction of 1st fragment-only (= broken or stalled) filaments that are visible in Figure 3 of Turner et al. 2012.

In this respect, we would like to emphasize that in our experimental setup, only a minimal fraction of the data points (<2%) resulted from broken or stalled filaments (see also Figure 7 and Figure 3—figure supplement 1 of our manuscript) and had no impact on the model fit.

2) In respect to the data analysis, Turner et al. 2012 did not exclude filaments that broke or stopped growing during their experiment because of the limitations of the 2-color approach. A large fraction of the analysed filaments of Turner et al. 2012 indeed appear to have stopped growing or were broken (compare Figure 3 of Turner et al. 2012 and Figure 7, where many filaments with only the basal fragment (= zero 2nd fragment growth) are apparent).

The authors did not exclude those filaments, in contrast to our approach, where we could readily identify and exclude them from our data analysis using a 3- or 6-color approach (Figure 7). We note that we observed only very few broken/stalled filaments in our data, likely due to the more physiological culture growth conditions.

3) While we do not know how to exactly convert the experimental conditions of Turner et al. 2012 quantitatively into our model, we performed a simulation of increasing fractions of broken filaments using our multi-colour data set. As shown in Figure 7 and the new Figure 3—figure supplement 2, the broken/stalled filaments accumulate on the x-axis and below the fit curve.

Accordingly, we conclude that a high fraction of broken/stalled filaments would mask the otherwise obvious length-dependency of filament growth and prevents an accurate fit. We believe that if broken/stalled filaments were excluded from Figure 3 of Turner et al. 2012, their data would very likely also reveal that growth rate decreases with increasing filament length.

Author response image 1.Comparison of the 2-color filament growth rate measurement of Turner et al. 2012 with our multi-colour filament growth rate data.a. 2-color growth rate data reported in Figure 3 of Turner et al. 2012; (**A**) represents unsheared filaments and (**B**) represents data from sheared filaments. The high number of points on the x-axis suggests that many filaments are broken or stalled. b. Raw data of our multi-colour filament growth rate measurements. Upper left panel represents the raw multi-colour data reported in the present manuscript and highlights apparently stalled or broken filaments. Lower panel represents the basal-apical growth coordinates from the multi-colour data and highlights as red dots the data derived from apparently stalled or broken filaments. Importantly, those data points accumulate below the fit curve, which prevents an accurate fit when the proportion of broken/stalled filaments is too high. c. Simulation of random filament breakage (of probability *p_break_*) on our dataset demonstrate that data points accumulate on the x-axis and below the fit curve (in red). It is crucial to note that a high fraction of broken filaments masks the obvious length-dependency of filament growth and prevents an accurate fit on the complete set of points (linear fit in blue; compare with Figure 3 of Turner et al. 2012).**DOI:**
http://dx.doi.org/10.7554/eLife.23136.023

4) We note that our filament growth data report filaments ranging from 0.2–10 µm, which is a substantially wider range compared to the distribution of filaments from Turner et al. 2012 (primarily filaments of 4–6 µm length were measured). In this respect, Figure 5 of Turner et al. 2012 does not report the true spread of the data and thus the apparent linear fit is difficult to interpret.

5) The filament labelling protocol of Turner et al. 2012 further included a long-term incubation step overnight at 7 °C in contrast to the in situ labelling in our experimental setup (We added the maleimide dyes during normal culture growth to minimize perturbations. This procedure did not affect bacterial growth or the growth rate of flagellar filaments. See also Figure 2—figure supplement 1 of our manuscript). Further, Turner et al. 2012 used substantially higher maleimide dye concentrations, which might have had additional effects on culture growth (40×–100× more dye than we used during our experiments = 1 mM dye compared to 10-25 µM used by us).

It appears possible that long-term incubation of their samples at 7 °C overnight might have negatively affected the rate of filament elongation. The injection rate of flagellin subunits into the growing filament would be readily affected by altered protein translation efficiency and changes in the available proton motive force (compare also Figure 6 of our manuscript). Supporting this possibility, we also observed quasi-linear growth of the filament if the injection rate and not flagellin transport was rate-limiting.

We thus speculate that the long manipulations and storage of the cells at 7 °C might have affected the filament growth rate of a subpopulation of the samples of Turner et al. 2012 due to perturbations of the injection rate. Changes in the cell's available proton motive force could negatively affect the injection rate and increase the dispersion of the data. This is emphasised by a comparison of Figure 3 of Turner et al. 2012 with our growth rate experiment under conditions of reduced proton motive force (Figure 8 and new Figure 3—figure supplement 2).

Author response image 2.Filament growth rate under conditions of reduced proton motive force.a. Basal-apical length coordinates based on the 2-color filament length data reported in Figure 3 of Turner et al. 2012. b. Basal-apical length coordinates of our growth rate experiment under conditions of reduced proton motive force. The curve in red represents the observed filament growth under physiological proton motive force. We presume that the length data reported in Figure 3 of Turner et al. 2012 comprises sub-populations of bacteria with decreased proton motive force due to extensive experimental manipulations. Please note that the duration of apical fragment growth was 60 minutes in our experiment, in contrast to 90 minutes in Turner et al. 2012.**DOI:**
http://dx.doi.org/10.7554/eLife.23136.024

6) Finally, it is important to highlight that we obtained very similar length-dependent filament growth dynamics and growth rates using three independent experimental approaches – and only one is similar (but improved) to the approach used by Turner et al. 2012:

First, we used population-immunostaining of filaments, where we determined the length of filaments at the end of the experiment. Here, the bacteria grew normally in batch culture without any manipulations prior to fixation (Figure 1, Figure 2—figure supplement 1).

Second, we performed continuous flow immunostaining of individual filaments during normal culture growth (Figure 1, Figure 1—figure supplement 2). Again, the bacteria were not manipulated during filament assembly and filament growth was observed in real-time.

Third, we significantly improved the quantity and resolution of our filament growth data using a 3- and 6-color sequential labelling approach. As described above in detail, we performed the filament labelling in situ during normal culture growth in order to minimize potential perturbations (Figure 2, Figure 3).